# The VOLNA-OP2 Tsunami Code (Version 1.5)

Istvan Z Reguly[1], Daniel Giles[5], Devaraj Gopinathan[2], Laure Quivy[6], Joakim H Beck[3], Michael B Giles[4], Serge Guillas[2], and Frederic Dias[5]

[1]Pázmány Péter Catholic University, Faculty of Information Technology and Bionics, Prater u 50/a, 1088 Budapest, Hungary
[2]Department of Statistical Science, University College London, London, UK
[3]Computer, Electrical and Mathematical Science and Engineering Division (CEMSE), King Abdullah University of Science and Technology (KAUST), Thuwal, 23955-6900, Saudi Arabia
[4]Math Institute, University of Oxford, Oxford, UK
[5]School of Mathematics and Statistics, University College Dublin, Dublin, Ireland
[6]Centre de Mathématiques et de Leurs Applications (CMLA), Ecole Normale Supérieure, Paris-Saclay, Centre National de la Recherche Scientifique, Université Paris-Saclay, 94235 Cachan, France

**Correspondence:** Istvan Z Reguly (reguly.istvan@itk.ppke.hu)

**Abstract.** In this paper, we present the VOLNA-OP2 tsunami model and implementation; a finite volume non-linear shallow water equations (NSWE) solver built on the OP2 domain specific language (DSL) for unstructured mesh computations. VOLNA-OP2 is unique among tsunami solvers in its support for several high performance computing platforms: CPUs, the Intel Xeon Phi, and GPUs. This is achieved in a way that the scientific code is kept separate from various parallel implementations, enabling easy maintainability. It has already been used in production for several years, here we discuss how it can be integrated into various workflows, such as a statistical emulator. The scalability of the code is demonstrated on three supercomputers, built with classical Xeon CPUs, the Intel Xeon Phi, and NVIDIA P100 GPUs. VOLNA-OP2 shows an ability to deliver productivity to its users, as well as performance and portability across a number of platforms.

## 1 Introduction

After the Indian Ocean tsunami of 26 December 2004, Bernard et al. (2006) emphasized that one of the greatest contributions of science to society is to serve it purposefully, as when providing forecasts to allow communities to respond before a disaster strikes. In the last twelve years, the numerical modelling of tsunamis has experienced great progress (Behrens and Dias (2015)). There is a variety of mathematical models, such as the shallow-water equations (Titov and Gonzalez (1997); Liu et al. (1998); Gailler et al. (2013); Zhang and Baptista (2008); Macías et al. (2017); Dutykh et al. (2011)), the Boussinesq equations (Kennedy et al. (2000); Lynett et al. (2002)), or the 3D Navier-Stokes equations (Abadie et al. (2012); Gisler et al. (2006)), and a large number of implementations, primarily for individual target computer architectures. The use cases of such models are wide ranging, and most rely on high numerical accuracy as well as high computational performance to deliver results - examples

include sensitivity analysis by Goda et al. (2014), probabilistic tsunami hazard assessments by Geist and Parsons (2006); Davies et al. (2017); Anita et al., and more efficient and informed tsunami early warning by Yusuke et al.; Castro et al. (2015).

For widespread use three key ingredients are needed; first, the stability and robustness of the numerical approach, that gives a confidence in the results produced, second, the computational performance of the code, which allows for getting the right results quickly, efficiently utilising the available computational resources, and third, the ability to integrate into a workflow, allowing for simple pre- and post-processing, efficiently supporting the kinds of use cases that come up - for example large numbers of different initial conditions.

In the Related Work section we discuss a number of codes currently being used in production, and as such are trusted and reliable codes, already being used as part of a workflow. Yet, the computational performance of most of these codes is "good enough"; they were written by domain scientists, and may have been tuned to one architecture or an other, but for example, GPU support is almost non-existent. In today's and tomorrow's quickly changing hardware landscape however, "future-proofing" numerical codes is of exceptional importance for continued scientific delivery. Domain scientists can not be expected to keep up with architectural advances, and spend a significant amount of time re-factoring code to new hardware. *What* to compute must be separated from *how* it is computed - indeed in a recent paper by Lawrence et al. (2017), leaders in the weather community chart the ways forward, and point to Domain Specific Languages (DSLs) as a potential way to address this issue.

OP2, by Mudalige et al. (2012), is such a DSL, embedded in C/C++ and Fortran; it has been in development since 2009: it provides an abstraction for expressing unstructured mesh computations at a high-level, and then provides automated tools to translate scientific code written once, into a range of high-performance implementations targeting multi-core CPUs, GPUs, and large heterogeneous supercomputers. The original VOLNA model (Dutykh et al. (2011)) was already discussed and validated in detail - it was used in production for small-scale experiments and modelling, but was inadequate for targeting large-scale scenarios and statistical analysis, therefore it was re-implemented on top of OP2; this paper describes the process, challenges and results from that work.

As VOLNA-OP2 delivered a qualitative leap in terms of possible uses due to the high performance it can deliver on a variety of hardware architectures, its users have started integrating it into a wide variety of workflows; one of the key uses is for uncertainty quantification; for the stochastic inversion problem of the 2004 Sumatra tsunami in Gopinathan et al. (2017), for developing Gaussian process emulators which help reduce the number of simulation runs (Beck and Guillas (2016); Liu and Guillas (2017)), applications of stochastic emulators to a submarine slide at the Rockall Bank (Salmanidou et al. (2017)), a study of run-up behind islands (Stefanakis et al. (2014)), the durability of oscillating wave surge converters when hit by tsunamis (O'Brien et al. (2015)), tsunamis in the St. Lawrence estuary (Poncet et al. (2010)), a study of the generation and inundation phases of tsunamis (Dias et al. (2014)), and others.

The time-dependency in the deformation enables the tsunami to be actively generated (Dutykh and Dias (2009)). This is a step-forward from the common passive mode of tsunamigenesis that utilises an instantaneous rupture. The active mode is particularly important for tsunamigenic earthquakes with long and slow ruptures, *e.g.* the 2004 Sumatra-Andaman event (Lay et al. (2005); Gopinathan et al. (2017)) and submerged landslides (Løvholt et al. (2015)), *e.g.* the Rockall Bank event (Salmanidou et al. (2017)).

These applications present a number of challenges in integration into the workflow, as well as scalable performance: the need for extracting snapshots of state variables on the full mesh, or at a number of specified locations, capturing the maximum wave elevation or inundation - all in the context of distributed memory execution.

As the above references indicate, VOLNA-OP2 has already been key in delivering scientific results in a range of scenarios, and through the collaboration of the authors, it is now capable of efficiently supporting a number of use cases, making it a versatile tool to the community, therefore we have now publicly released it: it is freely available at github.com/reguly/volna.

The rest of the paper is organised as follows: Section 2 discusses related work, Section 3 presents the OP2 library, upon which VOLNA-OP2 is built, Section 4 discusses the VOLNA simulator itself, its structure and features, Section 5 discusses performance and scalability results on CPUs and GPUs, and finally Section 6 draws conclusions.

## 2  Related Work

Tsunamis have long been a key target for scientific simulations. Behrens and Dias (2015) give a detailed look at various mathematical, numerical, and implementational approaches to past and current tsunami simulations. The most common set of equations solved are the shallow water equations, and most codes use structured and nested meshes. A popular discretisation is finite differences, such codes include: NOAA's MOST (Titov and Gonzalez (1997)), COMCOT (Liu et al. (1998)), CENALT (Gailler et al. (2013)). On more flexible meshes many use the finite element discretisation, such as SELFE (Zhang and Baptista (2008)) and TsunAWI (Harig et al. (2008)), ASCETE (Vater and Behrens (2014)), Firedrake-Fluids (Jacobs and Piggott (2015)) or the finite volume discretisation, such as the VOLNA code (Dutykh et al. (2011)), GeoClaw (George and LeVeque (2006)) or HySEA (Macías et al. (2017)). Another model is described by the Boussinesq equations - these equations and the solver are more complex than shallow-water solvers. Since they are primarily needed only for dispersion (Glimsdal et al. (2013)); they are used less commonly, examples include FUNWAVE Kennedy et al. (2000) and COULWAVE (Lynett et al. (2002)). Finally, the 3D Navier-Storkes equations provide the most complete description, but they are significantly more complex than other models - examples include SAGE (Gisler et al. (2006)) and the work of Abadie et al. (2012).

Most of these codes described above work on CPUs, and while there has been some work on GPU implementations by Satria et al. (2012); Liang et al. (2009a); Brodtkorb et al. (2010); Acuña and Aoki (2009); Liang et al. (2009b), which use structured meshes and finite differences or finite volumes, it is unclear whether these are used in production, and they are not open source. Celeris Tavakkol and Lynett (2017) is a Boussinesq solver that uses finite volumes and a structured mesh - it is hand-coded for GPUs using graphics shaders, and its source code is available, however it can only use a single GPU.

As far as we are aware, only Tsunami-HySEA (Macías et al. (2017)), also using finite volumes, is using GPU clusters in production - that code however only supports GPUs, and is hand-written in CUDA. Performance reported by Castro et al. (2015) on a 10M point testcase shows a strong scaling efficiency going from 1 GPU to 12 GPUs between 88% and 73% (overall 12 GPUs are 5.88 faster than 1 GPU), and a 25× speedup with 1 GPU over an unspecified (likely single core) CPU implementation. Direct comparison to VOLNA-OP2 is not possible since Tsunami-HySEA uses (nested) structured meshes, and the multi-GPU version is not open source.

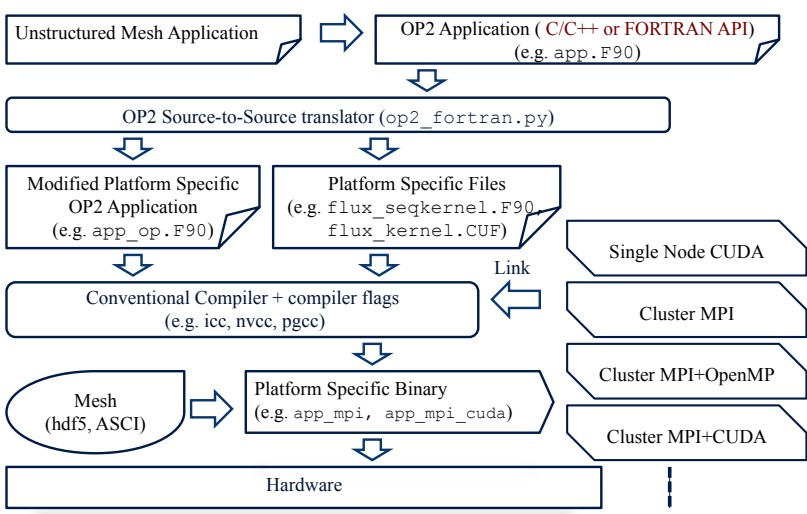

**Figure 1.** Build system with OP2

## 3 The OP2 Domain Specific Language

The OP2 library (Mudalige et al. (2012)) is a domain specific language embedded in C and Fortran that allows unstructured mesh algorithms to be expressed at a high level, and provides automatic parallelisation and a number of other features. It provides an abstraction that lets the domain scientist describe a mesh using a number of sets (such as quadrilaterals or vertices),

connections between these sets (such as edges-to-nodes), and data defined on sets (such as $x, y$ coordinates on vertices). Once the mesh is defined, an algorithm can be implemented as a sequence of parallel loops, each over all elements of a given set applying different "kernel functions", accessing data either directly on the iteration set, or indirectly through at most one level of indirection. This abstraction enables the implementation of a wide range of algorithms, such as the finite volume algorithms that VOLNA uses, but it does require that for any given parallel loop, the order of execution must not affect the end result

(within machine precision) - this precludes the implementation of e.g. Gauss-Seidel iterations.

OP2 enables its users to write an application only once using its API, which is then automatically parallelised to utilise multi-core CPUs, GPUs, and large supercomputers through the use of MPI, OpenMP and CUDA. This is done in part through a code generator that parses the parallel loop expressions and generates code around the computational kernel to facilitate parallelism and data movement, and in part through different back-end libraries that manage data, including MPI halo exchanges, or GPU

memory management, as shown in Figure 1. For more details see Giles et al. (2011); Mudalige et al. (2012).

### 3.1 Parallelisation Approaches in OP2

OP2 takes full responsibility for orchestrating parallelism and data movement - from the user perspective, the code written looks and feels like sequential C code that makes calls to an external library. To utilise clusters and supercomputers, OP2 uses the Message Passing Interface (MPI) to parallelise in a distributed memory environment; once the mesh is defined by

the user, OP2 automatically partitions and distributes it among the available resources. It uses the standard owner-compute

approach with halo exchanges, and overlaps computations with communications. In conjunction with MPI, OP2 uses a number of shared-memory parallelisation approaches, such as CUDA and OpenMP.

A key challenge in the fine-grained parallelisation of unstructured mesh algorithms is the avoidance of race conditions when data is indirectly modified. For example, in a parallel loop over edges, when indirectly incrementing data on vertices, multiple edges may try to increment the same vertex, leading to race conditions. OP2 uses a colouring approach to resolve this; elements of the iteration set are grouped into mini-partitions, and each element within these mini-partitions is coloured, so no two elements of the same colour access the same value indirectly. Subsequently mini-partitions are coloured as well. For CUDA, we assign mini-partitions of the same colour to different CUDA thread blocks, and for OpenMP to different threads. There is then a global synchronisation between different mini-partition colours. In case of CUDA, threads processing elements within each thread block use the first level of colouring to apply increments in a safe way, with block-level synchronisation in-between. Code generation that is suitable for auto-vectorisation by the compilers is also supported; it carries out the packing and unpacking of vector registers. Previous work describes further details and performance comparisons on various architectures, these are available in Mudalige et al. (2012); Reguly et al. (2007).

### 3.2 Input and Output

OP2 supports parallel file I/O through the HDF5 library (The HDF Group (2000-2010)), which is critically important to its integration into VOLNA's workflow: reading in the input problem and writing out data required for analysis simultaneously on multiple processes.

## 4 The VOLNA simulator

### 4.1 Model, numerics, previous validation

The finite volume (FV) framework is the most natural numerical method to solve the non-linear shallow water equations (NSWE), in part because of their ability to treat shocks and breaking waves. It belongs to a class of discretisation schemes that are highly efficient in the numerical solution of systems of conservation laws, which are common in compressible and incompressible fluid dynamics. Finite Volume methods are preferred over finite differences and often over finite elements because they intrinsically address conservation issues, improving their robustness: total energy, momentum and mass quantities are conserved exactly, assuming no source terms, and appropriate boundary conditions. The code was validated against the classical benchmarks in the tsunami community as described below.

### 4.2 Numerical model

Following the needs of the target applications, the following non-dispersive NSWEs (in Cartesian coordinates) form the physical model of VOLNA:

$$H_t + \nabla \cdot (H\boldsymbol{v}) = 0 \tag{1}$$

$$(H\boldsymbol{v})_t + \nabla \cdot \left( H\boldsymbol{v} \otimes \boldsymbol{v} + \frac{g}{2} H^2 \boldsymbol{I}_2 \right) = gH\nabla d \tag{2}$$

Here, $d(\boldsymbol{x},t)$ is the time-dependent bathymetry, $\boldsymbol{v}(\boldsymbol{x},t)$ is the horizontal component of the depth-averaged velocity, $g$ is the acceleration due to gravity and $H(\boldsymbol{x},t)$ is the total water depth. Further, $\boldsymbol{I}_2$ is the identity matrix of order 2. The tsunami wave height or elevation of free surface $\eta(\boldsymbol{x},t)$, is computed as,

$$\eta(\boldsymbol{x},t) = H(\boldsymbol{x},t) - d(\boldsymbol{x},t) \tag{3}$$

where the sum of static bathymetry $d_s(\boldsymbol{x})$ and the dynamic seabed uplift $u_z(\boldsymbol{x},t)$ constitute the dynamic bathymetry,

$$d(\boldsymbol{x},t) = d_s(\boldsymbol{x}) + u_z(\boldsymbol{x},t) \tag{4}$$

$d_s$ is usually sourced from bathymetry datasets pertaining to the region of interest (say, global datasets like ETOPO1/GEBCO or regional bathymetries). The vertical component $u_z(\boldsymbol{x},t)$ of the seabed deformation is calculated depending on the physics of tsunami generation, *e.g.* via co-seismic displacement for finite fault segmentations by Gopinathan et al. (2017), submarine sliding by Salmanidou et al. (2017, 2018) *etc.*.

In addition to the capabilities of employing active generation and consequent tsunami propagation, VOLNA also models the run-up/run-down (*i.e.* the final inundation stage of the tsunami). These three functionalities qualify VOLNA to simulate the entire tsunami life-cycle. The ability of the NSWEs (1-2) to model both propagation, as well as run-up and run-down processes was validated in Kervella et al. (2007) and Dutykh et al. (2011), respectively. Thus, the use of uniform model for the entire life-cycle obviates many technical issues such as the coupling between the sea bed deformation and the sea surface deformation and the use of nested grids.

VOLNA uses the cell-centered approach for control volume tesselation, meaning that degrees of freedom are associated with cell barycenters. However, in order to improve the spatial accuracy a second order extension is employed. A local gradient of the physical variables over each cell is calculated, then a limited linear projection of the variables at the cell interfaces is used within the numerical flux solver. The limiter used is a restrictive version of the scheme purposed by Barth and Jespersen (1989), the minimum calculated limiter of the physical variables within a cell is used in the reconstruction, this limiter ensures that numerical oscillations are constrained in realistic cases. A Harten-Lax-van Leer (HLLC) numerical flux which incorporates the contact discontinuity is used to ensure that the standard conservation and consistency properties are satisfied: the fluxes from adjacent triangles that share an edge exactly cancel when summed and the numerical flux with identical state arguments reduces to the true flux of the same state. Details of the numerical implementation can be found in Dutykh et al. (2011).

### 4.3 Validation

The original version of VOLNA was thoroughly validated against the National Tsunami Hazard Mitigation Program (NTHMP) benchmark problems Dutykh et al. (2011). A brief look at how the new implementation, which utilizes the more restrictive limiter performs with regards to two benchmark problems is given below. The reader is referred to the original paper Dutykh et al. (2011) or the NTHMP website for further details on the set-up of the benchmark problems.

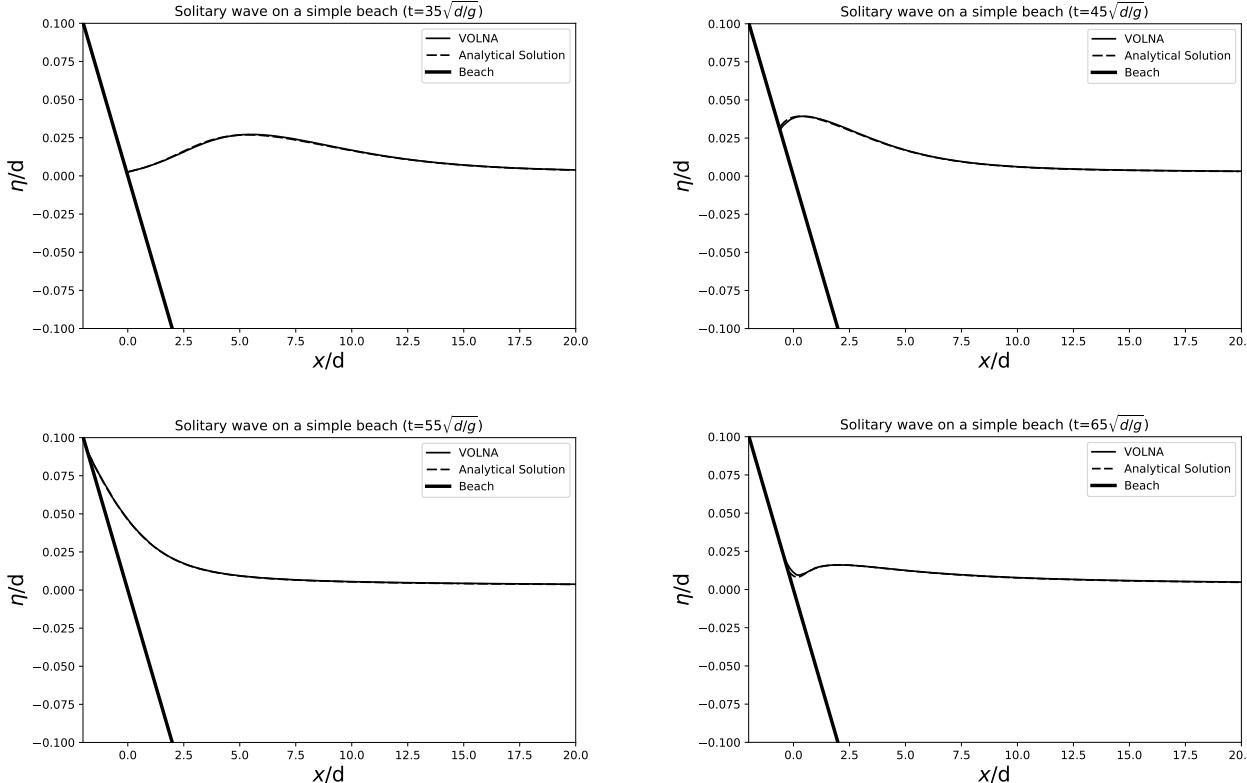

**Figure 2.** Solitary wave on a simple beach - Comparison between the simulated run-up and analytical solution at the shoreline (Time = 35, 45, 55, 65 $\sqrt{d/g}$). Solid line - VOLNA, Dashed line - Analytical Solution, Thick line - Beach

### 4.3.1 Benchmark Problem 1 - Solitary Wave on a Simple Beach

The analytical solution to the run up of a solitary wave on a sloping beach was derived by Synolakis (1987). Thus, in this benchmark problem one compares the simulated results with the derived analytical solution.

**Set Up**

The beach bathymetry comprises of a constant depth ($d$) followed by a sloping plane beach of angle $\beta = \text{arccot}(19.85)$. The initial water level is defined as a solitary wave of height $\eta$ centered at a distance $L$ from the toe of the beach and the initial wave-particle velocity is proportional to the initial water level.

$$H(x,0) = \eta\text{sech}^2(\gamma(x - X_1)/d) \tag{5}$$

$$u(x,0) = -\sqrt{\frac{g}{d}}H(x,0) \tag{6}$$

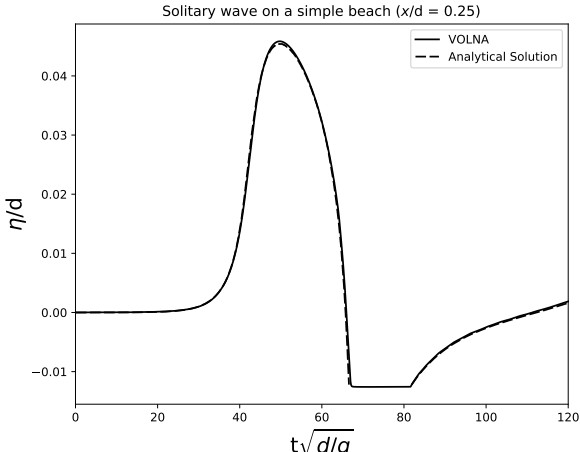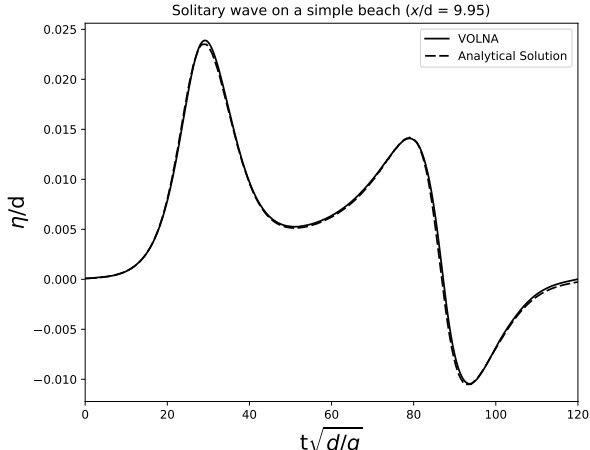

**Figure 3.** Solitary wave on a simple beach - Comparison between VOLNA and solution at different locations: (a) $x/d = 0.25$: Notice that the location becomes 'dry' for t $\approx (67\sqrt{d/g}) - 82\sqrt{d/g}))$, (b) $x/d = 9.95$.

where $x = X_0 = d\cot(\beta)$, $L = \text{arccosh}(\sqrt{20})/\gamma$, $X_1 = X_0 + L$, and $\gamma = \sqrt{3\eta/4d}$. For this benchmark problem the following ratio must also hold: $\eta/d = 0.019$.

**Tasks**

In order to verify the model, the wave run up at various time steps (Figure 2) and the wave height at two locations ($x/d = 0.25$ and $x/d = 9.95$) (Figure 3) are compared to the analytical solution.

It can be seen from the plots above that the agreement between numerical results and the analytical solutions is very good. So therefore, the new implementation of the model is able to accurately simulate the run-up of the solitary wave.

### 4.3.2 Benchmark Problem 2 - Wave Run-Up onto a Complex 3D Beach

This benchmark problem involves the comparison of laboratory results for a tsunami run up onto a complex 3D beach with simulated results. The laboratory experiment reproduces the 1993 Hokkaido-Nansei-Oki tsunami which struck the Island of Okushiri, Japan. The experiment is a 1:400 scale model of the bathymetry and topography around a narrow gully and the tsunami is an incident wave fed in as a boundary condition.

**Set Up**

The computational and laboratory domain corresponds to a 5.49m by 3.40m wave tank and the bathymetry for the domain is given for 0.014m by 0.014m grid cells. The incoming wave is incident on the $x$=0m boundary and is defined for the first 22.5s (Figure 4(a)), after which it is recommended that a non-reflective boundary condition be set. At $y$=0, $y$= 3.4 and $x$=5.5m fully reflective boundaries are to be defined.

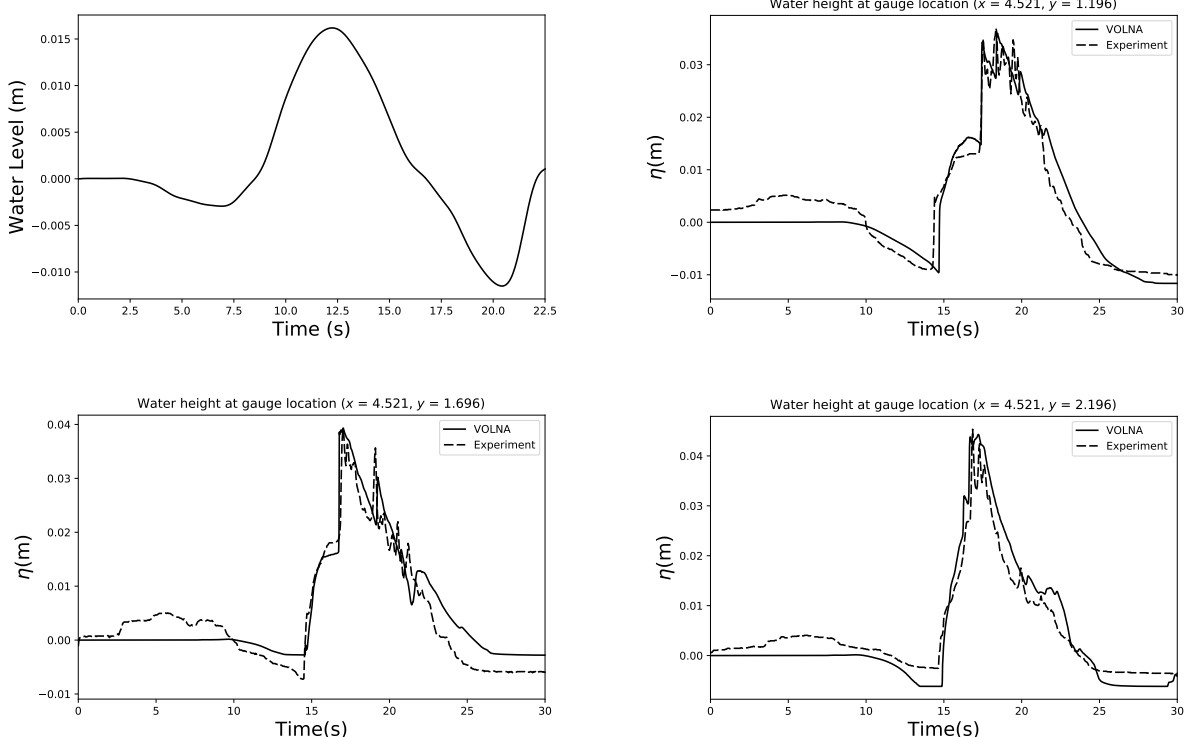

**Figure 4.** Benchmark Problem 2: (a) The incoming water level incident on the $x$=0m boundary, Comparison between VOLNA and laboratory results at different locations: (b) $x$=4.521, $y$=1.196, (c) $x$=4.521, $y$=1.696, (d) $x$=4.521, $y$=2.196

### Tasks

The validation of the model involves comparing the temporal variation of the moving shoreline, the water height at fixed gauges and the maximum run up. For the basis of this brief validation, we compared the water height at three gauges installed in the tank, located at (4.521, 1.196), (4.521, 1.696) and (4.521, 2.196).

It can be seen from the gauge plots on Figure 4(b-d) that the first elevation wave arrives between 15 and 25s. The overall dynamics of this elevation wave is accurately captured by the model at all the gauges, particularly the arrival time and initial amplitude. Considering the results of the two benchmark tests and the full validation of the original VOLNA code, one can see that the new implementation which implements a more restrictive limiter still preforms satisfactorily and is consistent with the previous version.

### 4.4 Code structure

The structure of the code is outlined in Algorithm 1; the user inputs a configuration file (.vln), which specifies the mesh to be read in from *gmsh* files, as well as initial/boundary conditions of state variables, such as the bathymetry deformation starting the tsunami, which can be defined in various ways (mathematical expressions or files, or a mix of both). We use a variable timestep third-order (four stage) Runge-Kutta method for evolving the solution in time. In each iteration, events may be triggered;

e.g. further bathymetry deformations, displaying the current simulation time, or outputting simulation data to VTK files for visualisation.

---
**Algorithm 1** Code structure of VOLNA
***
    Initalise mesh from gmsh file

    Initialise state variables

    **while** $t < t_{final}$ **do**

        Perform pre-iteration events

        Third-order Runge-Kutta time stepper

            Determine local gradients of state variables on each cell

            Compute a local limiter on each cell

            Reconstruct state variables, compute boundary conditions and determine fluxes across cell faces

            Compute timestep

            Apply fluxes and bathymetric source terms to state variables on cells

        Perform post-iteration events

    **end while**
***

The original VOLNA source code was implemented in C++, utilising libraries such as Boost (Schling (2011)). This gives a very clear structure, abstracting data management, event handling and low level array operations for the higher level algorithm - an example is shown in Figure 5. While this coding style was good for readability, it had its limitations in terms of performance: there was excessive amounts of data movement and certain operations could not be parallelised - indirect increments with potential race conditions in particular. Some features - such as describing the bathymetry lift with a mathematical formula - were implemented with functionality and simplicity, not performance, in mind.

To better support performance and scalability, and thus allow for large-scale simulations, we have re-engineered the VOLNA code to use OP2 - the overall code structure is kept similar, but matters of data management and parallelism are now entrusted to OP2. To support parallel execution we separated the pre-processing step from the main body of the simulation: first the mesh and simulation parameters are parsed into a HDF5 data file, which can then be read in parallel by the main simulation, which also uses HDF5's parallel file I/O to write results to disk.

Performance-critical parts of the code, essentially any operations on the computational mesh, are re-implemented using OP2: they are written with an element-centric approach and grouped for maximal data reuse. Calculations that were previously a sequence of operations, each calculating all partial results for the entire mesh, now apply only to single elements (such as cells or edges), and OP2 automatically applies these computations to each element - this avoids the use of several temporaries and improves computational density. This process involves outlining the computational "kernel" to be applied at each set element (cell or edge) to a separate function, and writing a call to the OP2 library - a matching code snippet is shown in Figure 5.

The workflow of VOLNA is made of a few sources of information being created and given as inputs to the code. The first is the merged bathymetry and topography over the whole computational domain, i.e. the seafloor and land elevations, over which the flow will propagate. This is given through an unstructured triangular mesh. This is then transformed into a usable

```
outConservative.H  *= dt;                          inline void EvolveValuesRK2_2(const float *dT,
outConservative.U  *= dt;                                                        float *outConservative,
outConservative.V  *= dt;                                                        const float *inConservative,
                                                                                 const float *midPointConservative,
outConservative.H  += MidPointConservative.H;                                    float *out)
outConservative.U  += MidPointConservative.U;      {
outConservative.V  += MidPointConservative.V;        outConservative[0]  *= (*dT);
                                                     outConservative[1]  *= (*dT);
outConservative.H  += inConservative.H;              outConservative[2]  *= (*dT);
outConservative.U  += inConservative.U;
outConservative.V  += inConservative.V;              outConservative[0]  += midPointConservative[0];
                                                     outConservative[1]  += midPointConservative[1];
outConservative.H  *= .5;                            outConservative[2]  += midPointConservative[2];
outConservative.U  *= .5;
outConservative.V  *= .5;                            outConservative[0]  += inConservative[0];
                                                     outConservative[1]  += inConservative[1];
outConservative.H  =                                 outConservative[2]  += inConservative[2];
    ( outConservative.H.cwise()  <= EPS )
    .select( EPS, outConservative.H );               outConservative[0]  *= 0.5f;
outConservative.Zb = inConservative.Zb;              outConservative[1]  *= 0.5f;
ToPhysicalVariables( outConservative, out );         outConservative[2]  *= 0.5f;
//Implementation of ToPhysicalVariables:
ScalarValue  TruncatedH =                            outConservative[0]  = MAX(outConservative[0],EPS);
    ( outConservative.H.cwise()  < EPS )             outConservative[3]  = inConservative[3];
    .select( EPS, outConservative.H );
out.H = outConservative.H;                           //call to ToPhysicalVariables inlined
out.U = outConservative.U.cwise()  / TruncatedH;     float TruncatedH = outConservative[0];
out.V = outConservative.V.cwise()  / TruncatedH;     out[0] = outConservative[0];
out.Zb = outConservative.Zb;                         out[1] = outConservative[1] / TruncatedH;
                                                     out[2] = outConservative[2] / TruncatedH;
                                                     out[3] = outConservative[3];
                                                   }
                                                   ...
                                                   op_par_loop(EvolveValuesRK2_2, "EvolveValuesRK2_2", cells,
                                                     op_arg_gbl(&dT,1,"float", OP_READ),
                                                     op_arg_dat(outConservative,-1,OP_ID,4,"float",OP_RW),
                                                     op_arg_dat(inConservative,-1,OP_ID,4,"float",OP_READ),
                                                     op_arg_dat(midPointConservative,-1, OP_ID,4,"float",OP_READ),
                                                     op_arg_dat(values_new,-1,OP_ID,4,"float", OP_WRITE));
```

**Figure 5.** Code snippets from the original and OP2 versions

input to VOLNA via the *volna2hdf5* code to generate compact HDF5 files. The mesh is also renumbered with the Gibbs-Poole-Stokmeyer algorithm to improve locality.

The second is the dynamic source of the tsunami. It can be an earthquake or a landslide. To describe the temporal evolution of seabed deformation, either a function can be used, or a series of files. When a series of files is used (typically when another numerical model provides the spatio-temporal information of a complex deformation), there is a need to define the frequency of these updates in the so-called *vln* generic input file to VOLNA. A recent improvement has been the ability to define these series of files for a sub-region of the computational domain, and at possibly lower resolution. Performance is better when using a function for the seabed deformation, since I/O requirements for files can generate large overheads - VOLNA-OP2 allows for describing the initial bathymetry with an input file, and then specifying relative deformations using arbitrary code that is a function of spatial coordinates and time. Similarly, one can also define initial conditions for wave elevation and velocity.

The generic input file of VOLNA, includes information about the frequency of the updates in the seabed deformation, the virtual gauges where time series of outputs will be produced and possibly some options to output time series of outputs over the whole computational domain in order to create movies for instance. These I/O requirements obviously affect performance: the more data to output and the slower the file system, the larger the effect.

To simulate tsunami hazard for a large number of scenarios is computationally expensive, so VOLNA has been replaced in past studies by a statistical emulator, i.e. a cheap surrogate model of the simulator. To build the emulator, input parameters

are varied in a design of experiments, and the runs are submitted with these inputs to collect input-output relationships. The output of interest could for example be the waveforms, free surface elevation, and velocity, among others. The increase in flexibility in the definition of the region over which the earthquake source of the tsunami is defined reduces the size of the series of files used as inputs: this is really helpful when a set of simulations needs to be run. Similarly, the ability to specify the relative deformation using an arbitrary code that is a function of spatial coordinates and time also reduces the computational and memory overheads when running a set of simulations.

## 5 Results

### 5.1 Running VOLNA

A key goal of this paper is to demonstrate that by utilising the OP2 library, VOLNA delivers scalable high performance on a number of common systems. Therefore we take a testcase simulating tsunami propagation in the Indian Ocean, and run it on three different machines: NVIDIA P100 Graphical Processing Units (The Wilkes2 machine in Cambridge's CSD3), a classical CPU architecture in the Peta5-Skylake part of CSD3 (specifically dual-socket Intel Xeon Gold 6142 16-core Skylake CPUs), and Intel's Xeon Phi platform in Peta5-KNL (64-core Knights Landing-generation chips, configured in cache mode).

There are five key computational stages that make up 90% of the total runtime: a stage evolving time using the third-order Runge-Kutta scheme (*RK*), *gradients* computes gradients between cells a stage that computes the fluxes across the edges of the mesh (*fluxes*), a stage that computes the minimum timestep (*dT*), and a stage that applies the fluxes to the cell-centered state variables (*applyFluxes*). Each of these stages consist of multiple steps, but for performance analysis we study them in groups.

The *RK* stage is computationally fairly simple, no indirect accesses are made, cell-centered state variables are updated using other cell-centered state variables, and therefore parallelism is easy to exploit, and the limiting factor to performance will be the speed at which we can move data; achieved bandwidth. Both the *gradients* and the *fluxes* stages are computationally complex, and involve accessing large amounts of data indirectly through cells-to-cells and edges-to-cells mappings. The *dT* stage moves significant amounts of data to compute the appropriate timestep for each cell, triggering an MPI halo exchange as well, and then carries out a global reduction to calculate the minimum - particularly over MPI this can be an expensive operation, but overall it is limited by bandwdith. The *applyFluxes* stage, while computationally simple, is complex due to its indirect increment access patterns; per-edge values have to be added onto cell-centered values, and in parallelising this operation, OP2 needs to make sure to avoid race conditions. The performance of this loop is limited by the irregular accesses and control throughout the hardware. For an in-depth study of individual computational loops and their performance we refer the reader to our previous work in Reguly et al. (2007).

### 5.2 Tsunami demonstration case

For performance and scaling analysis, we employ the Makran subduction zone as the tsunamigenic source for the numerical simulations. Our region of interest extends from $55°E$ to $79°E$ and from $6°N$ to $30°N$. The bathymetry (Fig. 6(a)) is obtained from GEBCO (www.gebco.net). The region of interest is projected about the center latitude (*i.e.* $18°N$) to form the rectangular

**Table 1.** Details of the non-uniform (NU) triangular meshes

| Mesh | Name | Vertices | Edges | Triangles | Source $\lambda$ | Mesh size at coast |
|------|------|----------|-------|-----------|------------------|--------------------|
| | | $n_V$ | $n_E$ | $n_T$ | $\lambda_0$ | $h_{min}$ |
| $NU_0$ | 53.7M | 26863692 | 80564925 | 53701234 | 12.5 km | 125 m |
| $NU_1$ | 13.8M | 6931758 | 20771822 | 13840065 | 25 km | 250 m |
| $NU_2$ | 3.6M | 1812073 | 5414155 | 3602083 | 50 km | 500 m |
| $NU_3$ | 0.95M | 485453 | 1435017 | 949565 | 100 km | 1000 m |

**Table 2.** Finite fault parameters of the 4-segment tsunamigenic earthquake source

| Segment | Length ($l$) | Down-dip width ($w$) | Longitude | Latitude | Depth | Strike | Dip | Rake |
|---------|--------------|----------------------|-----------|----------|-------|--------|-----|------|
| $i$ | ($km$) | ($km$) | (°) | (°) | ($km$) | (°) | (°) | (°) |
| 1 | 220 | 150 | 65.23 | 24.50 | 10 | 263 | 6 | 90 |
| 2 | 188 | 150 | 63.08 | 24.23 | 10 | 263 | 7 | 90 |
| 3 | 199 | 150 | 61.25 | 24.00 | 5 | 281 | 8 | 90 |
| 4 | 209 | 150 | 59.32 | 24.32 | 5 | 286 | 9 | 90 |

computational domain for VOLNA in Cartesian co-ordinates (Fig. 6(b)). This translates to a region of approximately $2500\,km \times 2700\,km$ in area. The calculation of the sea-floor deformation or uplift (assumed instantaneous) is modeled via the Okada solution (Okada (1992)). This deformation is generated by the earthquake source which is modeled as a 4-segment finite fault model (Table 2) with a uniform slip of $30\,m$. The non-uniform meshes for the simulation are generated using Gmsh (Geuzaine and Remacle (2009)). A simple strategy is used to generate these meshes. Using the dimensions of the finite fault earthquake sources ($l \times w$), an approximate source wavelength ($\lambda_0 < min(l, w)$) of the tsunami, and the ocean depth of the Makran trench ($d_0 \sim 3\,km$), we calculate the time period (T) of the wave as, $T = \dfrac{\lambda_0}{\sqrt{gd_0}}$. Next, assuming that the time period of the tsunami is same everywhere in the domain, we get for a depth $d_n$, $\dfrac{\lambda_n}{\sqrt{d_n}} = \dfrac{\lambda_0}{\sqrt{d_0}}$, which in turn relates the characteristic triangle (or element) length $h_n$ for depth $d_n$ as, $h_n = \dfrac{\lambda_0}{k}\sqrt{\dfrac{d_n}{d_0}}$, where $k = 10$. At the shore (*i.e.* $d = 0$), a minimum mesh size ($h_{min}$) is specified. Linear interpolation is carried out to further smoothen the mesh gradation. A combination of $\lambda_0$ and $h_{min}$ is used to generate a series of non-uniform meshes (Table 1 and Figure 7). We also fix the triangle size as $25\,km$ for regions that are deep inland. Finally, Figure 8 shows the tsunami waveforms at two virtual gauge locations. Simulated time is $21660\,s$ for all mesh sizes, however, for timed runs at different scales on different platforms we restrict this to $2000\,s$ to conserve computer time.

## 5.3 Performance and Scaling on classical CPUs

As the most commonly used architecture, we first evaluate performance on a classical CPUs in the Cambridge CSD3 super-computer: dual-socket Xeon Gold 6142 CPU, with 16 cores each, supporting the AVX512 instruction set. We test a plain MPI configuration (32 processes per node), as well as a hybrid MPI+OpenMP configuration, with 2 MPI processes per node (1 per socket), and 16 OpenMP threads each, with process and thread binding enabled.

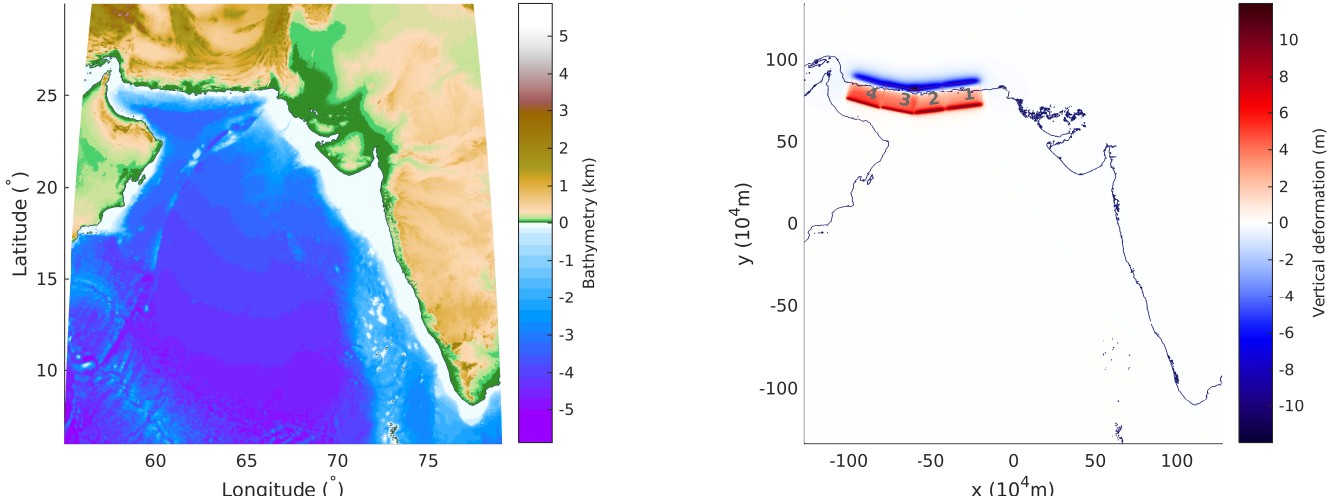

**Figure 6.** (a) Bathymetry from GEBCO's geodetic grid is mapped onto a Cartesian grid for use in VOLNA. (b) Uplift caused by a uniform slip of $30\,m$ in the 4 segment finite fault model (given in Table 2).

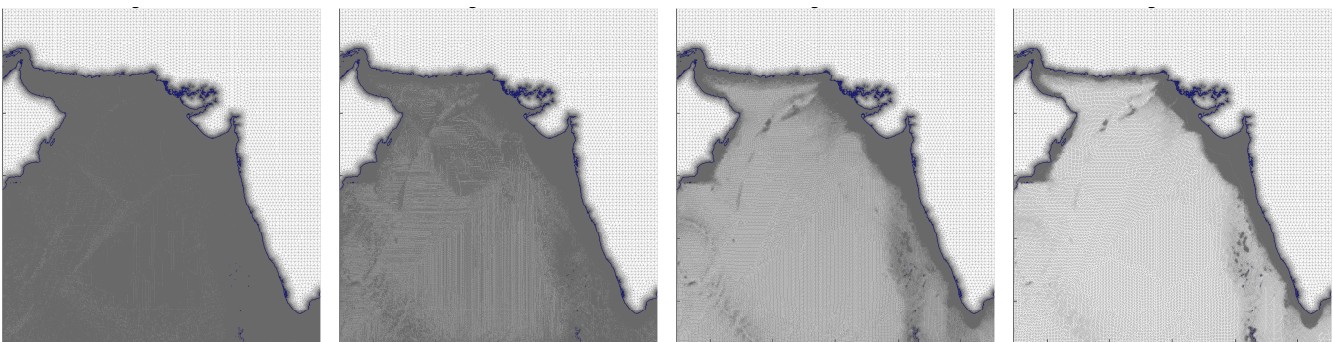

**Figure 7.** Non-uniform meshes corresponding to the test cases (see Table 1).

We use OP2's vectorised code generation capabilities, as described in Mudalige et al. (2016). The *RK* stage performs the same in both variants, however the *fluxes* and *dT* stages saw significant performance gains - the compiler did not automatically vectorise computations, it had to be forced to do so. The *applyFluxes* stage could not be vectorised due to a compiler issue.

On a single node with pure MPI, running the largest mesh, 9% of time was spent in the *RK* stage, achieving 182 GB/s throughput on average, 40% of time was spent in the *gradients* stage, achieving 108 GB/s, 25% of time was spent in the *fluxes* stage, achieving 142 GB/s, 12% of time was spent in the *dT* phase, achieving 65 GB/s, and 12% of time was spent in the *applyFluxes* stage, achieving 221 GB/s thanks to a high degree of data reuse. The maximum bandwidth on this platform is 189 GB/s as measured by STREAM Triad. The time spent in MPI communications ranged from 23% on the smallest mesh to 10% on the largest mesh.

When scaling to multiple nodes with pure MPI, as shown in Figure 9(a), it is particularly evident on the smallest problem that the problem size per node needs to remain reasonable, otherwise MPI communications will dominate the runtime: for the $NU_0$ mesh, at 32 nodes 251 seconds out of 308 total (81%). This can be characterised by the *strong scaling efficiency*; when

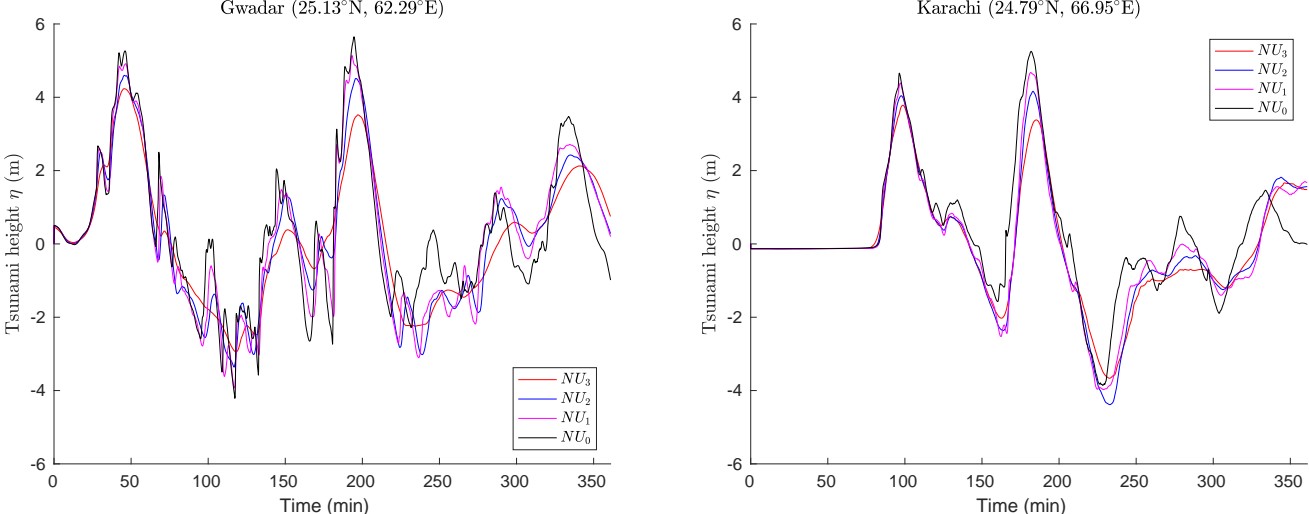

**Figure 8.** Tsunami waveforms at virtual gauges located at Gwadar and Karachi.

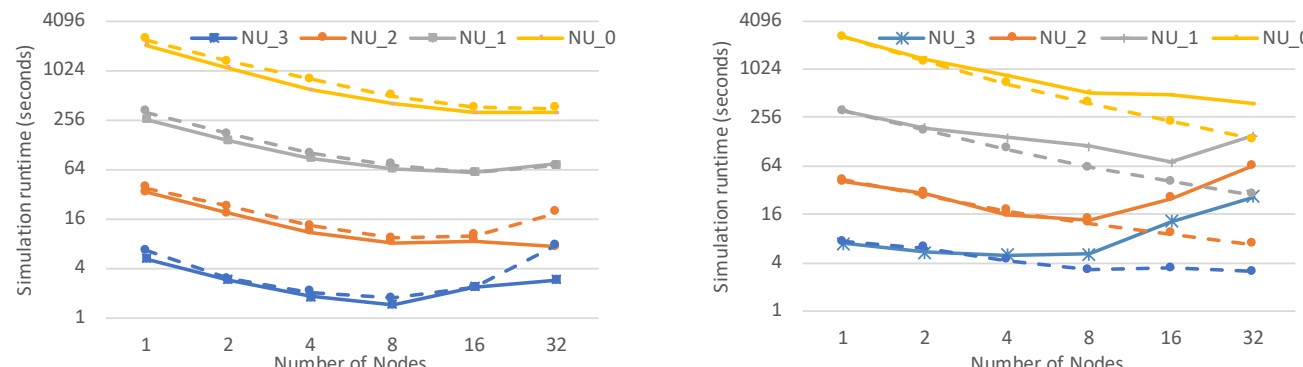

**Figure 9.** Performance scaling on (a) Peta5-CPU (Intel Xeon CPU) and (b) Peta5-KNL (Intel Xeon Phi) at different mesh sizes with pure MPI (solid) and MPI+OpenMP (dashed)

doubling the number of computational resources (nodes), what percentage of the ideal $2\times$ speedup is achieved. For small node counts these values remain above a reasonable 85%, but particularly for the smaller problems runtimes actually become worse. It is evident that on the Peta5-Skylake cluster the interconnect used for MPI communications becomes a bottleneck for scaling - this overhead is significantly lower on e.g. Archer on the largest mesh at 32 nodes it is only 32%.

5    We have also evaluated execution with a hybrid MPI+OpenMP approach, as shown with the dashed lines in Figure 9(a). However, on this platform it failed to outperform the pure MPI configuration.

### 5.4    Performance and Scaling on the Intel Xeon Phi

Second, we evaluate Intel's latest many-core chip, the Xeon Phi x7210, which integrates 64 cores, each equipped with AVX-512 vector processing units and supporting 4 threads, and built with a 16GB on-chip high-bandwidth memory, here used as

10    a cache for off-chip DDR4 memory. The chips were configured in the "quad" mode, and all 16GB as cache. We evaluate

a pure MPI approach (128 processes) as well as using 4 MPI processes, one per quadrant, and 32 OpenMP threads each. Bandwidth achieved as measured by STREAM Triad is 448 GB/s. Vectorisation on this platform is paramount for achieving high performance - every stage with the exception of *applyFluxes* was vectorised - the latter was not due to compiler issues.

On a single node with pure MPI, the straightforward computations of the *RK* stage can utilise the available high bandwidth very efficiently: only 8.3% of time spent here, achieving 194 GB/s. The *gradients* stage takes 42% of time, achieving 82 GB/s, the *fluxes* stage takes 25% of time and achieves 104 GB/s, *dT* 11.4% and achieves 46 GB/s, and the *applyFluxes* stage takes 11.6% and achieves 165 GB/s. On the largest mesh, it is 21% slower than a single node of the classical CPU system.

Performance when scaling to multiple nodes with pure MPI is shown in Figure 9(b): it is quite clear that scaling is worse than on the classical CPU architecture for smaller problem sizes - the Xeon Phi requires a considerably larger problem size per node to operate efficiently. Strong scaling efficiency is particularly poor on the smallest mesh, but even on the largest mesh it is only between 63-92%. Similarly to the classical CPU system, the interconnect becomes a bottleneck to scaling. Running with a hybrid MPI+OpenMP configuration on the Xeon Phi does improve scaling significantly, as shown in Figure 9(b) - this is due to having to exchange much fewer (but larger) messages. Strong scaling efficiency on the largest problem remains above 82%. At scale, at least on this cluster, the Xeon Phi can outperform the classical CPU system on a node-to-node basis of comparison.

## 5.5 Performance and Scaling on P100 GPUs

Third, we evaluate performance on GPUs - an architecture that has continually been increasing its market share in high performance computing thanks to its efficient parallel architecture. The P100 GPUs are hosted in the Wilkes2 system, with 4 GPU per node connected via the PCI-e bus. Each chip contains 60 Scalar Multiprocessors, with 64 CUDA cores each, giving a total of 3840 cores. There is also 16 GB of high-bandwidth memory on-package, with a bandwidth of 497 GB/s. To utilise these devices, we use CUDA code generated by OP2, and compiled with CUDA 9. Similarly to Intel's Xeon Phi, high vector efficiency is required for good performance on the GPU.

On a single GPU, running the second-largest mesh $NU_1$ (Because $NU_0$ does not fit in memory), 8.3% of runtime is spent in the *RK* stage, achieving 342 GB/s, *gradients* takes 50% if time, achieving only 136 GB/s due to its high complexity, *fluxes* takes 15%, achieving 379 GB/s thanks to a high degree of data re-use in indirect accesses, *dT* takes 4.4% and achieves 382 GB/s, and finally *applyFluxes* takes 20% of time, achieving 204 GB/s. Indeed, this last phase has the most irregular memory access patterns, which is commonly known to be degrading performance on GPUs. Nevertheless, even a single GPU outperforms a classical CPU node by a factor of 1.5, and the Xeon Phi by $1.85\times$.

Performance when scaling to multiple GPUs is shown in Figure 10; similarly to the Xeon Phi, GPUs are also sensitive to the problem size and the overhead of MPI communications, however given that there are 4 GPUs in one node, the overhead of communications is significantly lower. On the smallest problem efficiency drops from 78% to 58%, and on the largest problem from 95% to 89%. Thanks to much better scaling (due to lower MPI overhead), 32 GPUs are $6.9/2.6\times$ faster than 32 nodes of Xeon CPUs and Xeon Phis respectively.

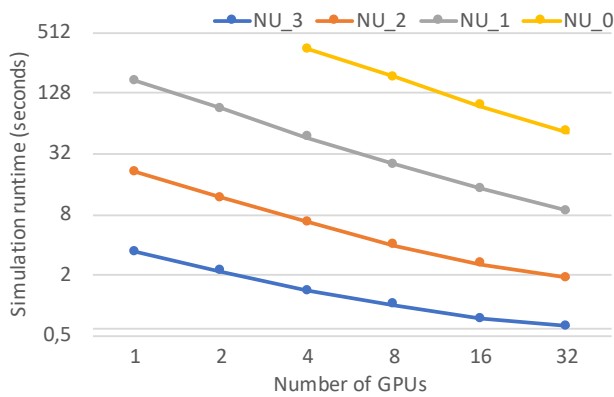

**Figure 10.** Performance scaling on Wilkes2 (P100 GPU) at different mesh sizes

### 5.6 Running Costs and Power Consumption

Ultimately, when one needs to decide what platform to run these simulations on, a key deciding factor aside from time-to-solution, is cost-to-solution. In the analysis above, aside from discussing absolute performance metrics, we have reported speedup numbers relative to other platforms - which from a performance benchmarking perspective is not strictly fair. However, such relative performance figures combined with the cost of access do help in the decision.

Admittedly the cost buying hardware, as well as the cost of core-hours or GPU-hours varies significantly, therefore here we do not look at specific prices. However, energy consumption is an indicator of pricing. A dual-socket CPU consumes up to 260 Watts, which is then roughly tripled when looking at the whole node, due to memory, disks, networking, etc. In comparison, the Intel Xeon Phi CPU has a TDP of 215 W, roughly 750 W for the node. A P100 GPU has a TDP of 300 W, but has to be hosted in a CPU system - the more GPUs in a single machine, the better amortised this cost is: the TDP of a GPU node in Wilkes2 is around 1.8 KW (4x250 for the GPUs, plus 800 for the rest of the system) - which averages to 450 W per GPU. Thus in terms of power efficiency GPUs are by far the best choice for VOLNA. Nevertheless, a key benefit of VOLNA-OP2, is that it can efficiently utilise any high performance hardware commonly available.

### 6 Conclusions

In this paper we have introduced and described the VOLNA-OP2 code; a tsunami simulator built on the OP2 library, enabling execution on CPUs, GPUs, and heterogeneous supercomputers. By building on OP2, the science code of VOLNA itself is written only once using a high-level abstraction, capturing *what* to compute, but not *how* to compute it. This approach enables OP2 to take control of the data structures and parallel execution; VOLNA is then automatically translated to use sequential execution, OpenMP, or CUDA, and by linking with the appropriate OP2 back-end library, these are then combined with MPI. This approach also future-proofs the science code: as new architectures come along, the developers of OP2 will update the back-ends and the code generators, allowing VOLNA to make use of them without further effort. This kind of ease-of-use and portability makes VOLNA-OP2 unique between the tsunami simulation codes. Through performance scaling and analysis of

the code on traditional CPU clusters, as well as GPUs and Intel's Xeon Phi, we have demonstrated that VOLNA-OP2 indeed delivers high performance on a variety of platforms and, depending on problem size, scales well to multiple nodes.

We have described the key features of VOLNA, the discretisation of the underlying physical model (*i.e.* NSWE) in the finite volume context and the third-order Runge-Kutta timestepper, as well as the input/output features that allow the integration of the simulation step into a larger workflow; initial conditions, and bathymetry in particular, can be specified in a number of ways to minimise I/O requirements, and parallel output is used to write out simulation data on the full mesh or specified points.

There is still a need for even more streamlined and efficient workflows. For instance, we could integrate within VOLNA, the finite fault source model for the slip with some assumptions on the rupture dynamics, we could also integrate the bathymetry-based meshing (the mesh needs to be tailored to the depth and gradients of the bathymetry to optimally reduce computational time). Indeed, there would be even less exchanges of files and more efficient computations, especially in the context of uncertainty quantification tasks such as emulation or inversion.

In the end, the gain in computational efficiency will allow higher resolution modelling, such as using $2\,m$ topography and bathymetry collected from LIDAR, i.e. a greater capability. It will allow greater capacity by enabling more simulations to be performed. Both of these enhancements will subsequently lead to better warnings more tailored to the actual impact on the coast as well as better urban planning since hazard maps will gain in precision geographically and probabilistically, due to the possibility of exploring a larger number of more realistic scenarios.

*Code availability.* The code is available at https://github.com/reguly/volna/, and DOI: 10.5281/zenodo.1413124 It depends on the OP2 library, which is also available at: https://github.com/OP-DSL/OP2-Common, and depends on an MPI distribution, parallel HDF5, and a partitioner, such as ParMetis or PT-Scotch. For GPU execution, the CUDA SDK and a compatible device is required.

*Competing interests.* The authors declare that they have no conflict of interest.

*Acknowledgements.* We would like to thank Endre László, formerly of PPCU ITK, who worked in the initial port of Volna to OP2. István Reguly was supported by the János Bólyai Research Scholarship of the Hungarian Academy of Sciences. Project no. PD 124905 has been implemented with the support provided from the National Research, Development and Innovation Fund of Hungary, financed under the PD_17 funding scheme. The authors would like to acknowledge the use of the University of Oxford Advanced Research Computing (ARC) facility in carrying out this work http://dx.doi.org/10.5281/zenodo.22558. Serge Guillas gratefully acknowledges support through the NERC grants PURE (Probability, Uncertainty and Risk in the Natural Environment) NE/J017434/1, and "A demonstration tsunami catastrophe risk model for the insurance industry" NE/L002752/1. Serge Guillas and Devaraj Gopinathan acknowledge support from the NERC project (NE/P016367/1) under the Global Challenges Research Fund: Building Resilience programme. Devaraj Gopinathan acknowledges support from the Royal Society, UK and Science and Engineering Research Board (SERB), India for the Royal Society-SERB Newton International Fellowship (NF151483). Daniel Giles acknowledges support by the Irish Research Council's Postgraduate Scholarship Programme.

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
