# Peer review of "The VOLNA-OP2 Tsunami Code (Version 1.5)"

_Geoscientific Model Development, 2018_

## Short Comment (SC1) · 12 Mar 2018

The precise version of the code discussed in the manuscript must be made available. The current best practice is for this code to be uploaded to a public repository and a DOI assigned. The DOI should be cited in the manuscript. github is inadequate because it does not readily link to the precise version of the code. However, making github code citable is not difficult; see: https://guides.github.com/activities/citable-code/

---

## Author Comment (AC1) · 12 Mar 2018

Apologies, we forgot to include this, the DOI for the code is: 10.5281/zenodo.1193675, which is being added to the Code Availability section of the paper.

---

## Referee Comment (RC1) · Anonymous Referee #1 · 29 Apr 2018

This paper present the development of a new generalized hardware version of the Volna tsunami code. The Volna code is one of the recognised Finite Volume codes in the tsunami community capable of simulation tsunami propagation and inundation over dry land. In this paper, the authors have developed a new version of the code, this time the code is embedded into a library (OP2) that allow the model to be compiled and run in different hardware configurations, including single CPU, multiple CPU's with Message Passing Interface, or Graphical Processing Unit (GPU) among others. The author's claim that is the first of its kind of such general interface tsunami models. To my knowledge also, there exists no other comparable tsunami models that are general in this sense. Therefore, the present study may possibly provide a good addition to the present tsunami literature and set of models. However, a set of amendments should

be undertaken before publication can be considered.

From my reading of the manuscript, the following main points are found:

This is presumably the first model implementation of its kind, and I find that the implementation should be of interest for the tsunami community, as well as other scientific communities with interest of solving shallow water wave equations or related problems without in depth knowledge of different types of hardware architecture. This part is highly regarded.

The study of the speedup on different types of hardware are also new, and the findings are interesting in their own right. However, if possible, I encourage the authors to see if it is possible to compare the model speedup also with other models (such as HySEA) for inter model comparison.

The validation of the model is entirely missing. I know that the previous VOLNA codes have been benchmarked towards NTHMP tests previously, but this is a new implementation. While one may expect a similar accuracy for this code as well, validation needs to be demonstrated. To emphasise this, the novelty of this paper actually hinges on some kind of proof; i.e. that the model can produce results consistent with previous versions. Moreover, no explicit tsunami results are shown, only results showing the speedup. As a minimum, some results showing that the tsunami code gives a reasonable output needs to be included. I would propose that the authors include one or two of the standard tsunami inundation benchmark tests. I'm sure the authors have some such tests available.

The text and reference list is a bit imbalanced with respect to the authors own work. It would be beneficial if some more external references are added, reference to external work is moved upfront, or alternatively, discussion of the authors own work that are not strictly relevant for this paper are omitted (some parts seems not strictly necessary, see below). In the line-by-line review below some examples are listed. The references in the related work section should be moved upfront (section 2 seems unnecessary).

Some edits to the text and references are needed. See the line by line comments below.

Line-by-line comments:

Page 1, line 14: More references to external work should preferably be placed up front (e.g. here). It makes sense to pay attention to the general literature first, and use this to put the authors own work into a general context thereafter.

Page 1, line 15: The statement "there are only a handful of codes that are suitable for integration into a workflow" is unsubstantiated, please remove.

Page 2, line 1: The science perspective is missing here, but is obvious, for instance the need for running sensitivity analysis (such as varibable slip or uncertainty assessments, e.g. Goda et al., 2014), probabilistic tsunami hazard assessments (e.g. Geist and Parsons, 2006; Davies et al., 2018; Grezio et al., 2017), or for more efficient and informed tsunami early warning (e.g. Oishi et al., 2015, Castro et al., 2015). I think it would strengthen the paper to mention and discuss such examples.

Page 2, lines 19-28: Reading this paragraph, you get the impression that the Volna code is unique with respect to workflow integration, which is not the case (see comment to Page 1, line 15). There are probably more than 10 codes worldwide that can do much of the same analysis. Granted, the new development presented in this paper provide new opportunities wrt hardware independence. This should be the main message.

Page 3: References and discussions in Related Work section should preferably be moved upfront.

Page 3, line 10: Unsubstantiated statement: "Since there is no consensus as to their advantage...". What do the authors mean here? Clarify, or remove statement. Simply, Boussinesq models are needed wherever tsunami dispersion is needed (see e.g. Glimsdal et al., 2013), otherwise the shallow water approximation is sufficient.

Page 3, line 17: The authors should provide a literature search here and add more

references, as GPU implementation of shallow water is under rapid development. As a minimum, the authors needs to add reference to the GPU SWE code by Brodtkorp et al. (2010) and the Boussinesq GPU code Celeris (Tavakkol and Lynett, 2017).

Page 4: I could not find anywhere from the discussion whether related OP2 applications have been performed for other (similar) applications of hyperbolic equation. If such implementations exists, it would be of interest to discuss their performance.

Page 5, line 9: Remove "very".

Page 5, lines 7-12: Treatments of shocks and breaking waves are probably the main reason FV are used, so this needs explicit mentioning.

Page 5, line 14: Replace "megatsunami" with either "large tsunami" or "transoceanic tsunami".

Page 5, line 15: Again, we refer to Glimsdal et al. (2013). Because frequency dispersion is a time dependent property, important of dispersion increases with time for a given initial condition, so it is not sufficient to refer to dispersion as weak just based on the properties at a given snapshot. The discussion here seems to merge the effect of dispersion on deep water waves and inundation, which are very different. Either the authors needs to clarify better, however, it would probably be better to omit this discussion here, and rather state that the present implementation is based on the non-linear shallow water model (you do not have to justify that dispersion is not included).

Page 6, first paragraph: This repeated reference to applications of the code seems awkward as it is not needed in this context, beside, this is already discussed in the introduction.

Page 9, lines 14-15: See previous comment.

Page 14, lines 29-31: I cant see that this is more relevant than other and more general applications such as PTHA and tsunami early warning. As said, a more general discussion with references from a broader literature is needed.

References:

Brodtkorb, A.R., Hagen, T.R., Lie, KA. et al. Comput. Visual Sci. (2010) 13: 341. https://doi.org/10.1007/s00791-010-0149-x

Castro, M.J., González-Vida, J.M., Macías, J., Ortega, S., and de la Asunción, M. (2015) Tsunami-HySEA: A GPU-based model for Tsunami Early Warning Systems, Proc XXIV Cong Diff Eq Appl, and XIV Cong Appl Math, Cádiz, June 8-12, 2015, 1–6

Davies G., Griffin, J., Løvholt, F., Glimsdal, S., Harbitz, C., Thio, H.K., Lorito, S., Basili, R., Selva, J., Geist, E., and Baptista, M.A. (2018), Geological Society, London, Special Publications, 456, 219-244, 23 February 2017, https://doi.org/10.1144/SP456.5

Geist E and Parsons T (2006), Probabilistic analysis of tsunami hazards, Nat Hazards, 37(3) 277-314

Glimsdal, S., Pedersen, G. K., Harbitz, C. B., and Løvholt, F. (2013), Dispersion of tsunamis: does it really matter?, Nat. Hazards Earth Syst. Sci., 13, 1507-1526, doi:10.5194/nhess-13-1507-2013

Goda K, Mai, P.M., Yasuda, T., and Mori, N. (2014), Sensitivity of tsunami wave profiles and inundation simulations to earthquake slip and fault geometry for the 2011 Tohoku earthquake, Earth, Planets and Space, 66: 105. https://doi.org/10.1186/1880-5981-66-105

Grezio, A., Babeyko, A., Baptista, M. A., Behrens, J., Costa, A., Davies, G.,. . . Thio, H. K. (2017). Probabilistic Tsunami Hazard Analysis: Multiple sources and global applications. Reviews of Geophysics, 55, 1158–1198. https://doi.org/10.1002/2017RG000579

Oishi, Y., Imamura, F. and Sugawara, D. (2015), Near-field tsunami inundation forecast using the parallel TUNAMI-N2 model: Application to the 2011 Tohoku-Oki earthquake combined with source inversions. Geophys. Res. Lett., 42: 1083-1091. doi: 10.1002/2014GL062577.

[Figure]

Tavakkol, S and Lynett, P (2017), Celeris: A GPU-accelerated open source software with a Boussinesq-type wave solver for real-time interactive simulation and visualization, Computer Physics Communications, 217, 117-127, ISSN 0010-4655,
* * *

---

## Referee Comment (RC2) · Anonymous Referee #2 · 11 Jul 2018

Summary:

This paper describes the design, implementation, and performance of VOLNA-OP2, a code for tsunami modelling based upon the shallow water equations solved by the finite volume method. VOLNA-OP2 is the result of long-standing interdisciplinary work, having the objective of porting the original C++ implementation, VOLNA (released in 2011), on top of OP2 (hence the name VOLNA-OP2), a framework for unstructured mesh applications. Thanks to the OP2 layer, VOLNA-OP2 can now run on a variety of computer architectures, including CPUs (e.g., Intel Xeon, Intel Xeon Phi) and GPUs. Support for both shared-memory parallelism (OpenMP, CUDA), distributed-memory parallelism (MPI), and SIMD vectorisation is provided. Therefore, there are two main benefits deriving from this work: a new codebase that drastically relieves the burden of extending

and maintaining VOLNA; higher performance as well as performance portability across architectures. These make VOLNA-OP2 an appealing candidate in the ecosystem of tsunami-modelling codes. An entire section discusses the performance achieved on several computer architectures.

Comment:

This work is a good example of what can be achieved through interdisciplinary research. VOLNA-OP2 overcomes the initial limitations of VOLNA by allowing the code to scale up to hundreds of cores and on different kinds of architecture, thus enabling science in new contexts and scenarios.

My questions, that should be addressed before publication, mostly concern the performance results section, as the rest already looks in solid shape (apart from some minor points listed below). Despite I understand that "showing how fast VOLNA runs on top of OP2" is not the main point of the article, I think that some of the following aspects should be clarified. This may also be helpful for people wanting to follow a code-modernisation approach similar to the one described in the submitted article.

* Use of uniform triangular meshes as opposed to non-uniform ones. Am I right if I say that the most typical use case of VOLNA-OP2 is with non-uniform meshes? If so, why using uniform ones? The mesh can have a drastic impact on the achieved performance, due to different MPI partitioning, load balancing, effectiveness of mesh renumbering etc. Is this a weakness of the analysis? if not, why?

* I know that with other OP2-based applications, traditionally, MPI+OpenMP has never really outperformed pure MPI (due to issues that have been extensively described in prior work), or at least not by a significant factor. Is the situation different with VOLNA? If so, can you say why? Speculation: is it because VOLNA is "more compute-bound" than other codes used in the past? I'm asking because I see that all numbers reported derive from MPI+OpenMP. By the way, I assume that you have taken care of pinning etc – can this be confirmed?

* Was data alignment enforced for directly accessed datasets ? I understand the same dataset can be accessed directly in one loop and indirectly in another loop, but perhaps there's hope to exploit some alignment in at least some of the memory-bounded loops?

* Can you be more precise as to why it was not possible to vectorise 'applyFluxes'? Can you share more details about how the vectorisation of the other loops has been achieved – is it via auto-vectorisation or something else?

* Which loops are compute-bound and which are memory-bound?

* Can you report the max memory bandwidth of *all* platforms? I think some are currently missing. Also, how was the memory bandwidth limit determined ? specs, STREAM, or...?

* Can you state the model of XeonPhi used? Some have 68 cores, not 64 like the one used in the experimentation.

* I think comparing the VOLNA-OP2 performance in different architectures is a bit unfair, at least for two reasons: 1- The number of "degrees-of-freedom" (or simply triangles) per core is generally different, and so is the proportion of time spent is computation and communication. Even data locality may have been impacted. 2- These architectures are profoundly different among them – in theoretical attainable peak performance, price, release date, etc. I don't like sentences such as "The GPU system with X1 nodes was Y times faster than the CPU system with X2 nodes"; I don't think they add much to the paper, and in fact they may be misleading. Perhaps you can normalise over a common metric, but again that's quite tough. Also, the Phi suffers a lot from the lack of the vectorisation in 'applyFluxes', and it's unclear whether this issue can be worked around or not.

* In the conclusions, I disagree with the sentence: "Through performance scaling and analysis of the code on (...), we have demonstrated that VOLNA-OP2 indeed delivers near-optimal performance on a variety of platforms (...)" . My point is that I don't think

that you have demonstrated that you're achieving near-optimal performance – it is true that some loops are relatively close to the architecture memory bandwidth limit, but others are not. We don't know the reason – in fact, we don't even know whether these loops are compute- or memory-bound (see point above). So I suggest to either drop that sentence, or rephrase it, or add a roofline plot.

* Of course, I assume that comparing the performance of VOLNA-OP2 to that of other codes is way too difficult, if not impossible

Code release and misc:

I see that version 1.0 has been "Zenodoized" – don't forget to add the DOI to the paper. Also, I suggest to move the code to its own organisation on GitHub so that permissions can be more easily set and contributing to VOLNA-OP2 gets easier. I also suggest to link VOLNA-OP2 from the 'apps' section of OP2.

On the language:

It might be just me, but it feels like that different sections of the paper have been written by different people. That is, the transition from some paragraphs to others is not as smooth as it might be. Also, there are some typos (e.g., outline -> outlined) and/or missing words ("the physical across") in various sections. All this should be improved prior to publication.

Minor notes:

* Section 3.1 might benefit from some figures. I know this is not the main point of the article, but I'm not sure that readers who are unfamiliar with OP2 will be able to understand how, for example, GPU parallelisation works. Maybe just cite some prior paper?

* In the performance section, Table 1 gives a name to three meshes: M1, M2, M3. These could be used systematically throughout the whole section, and in the plots as well, instead of referring to "the 1.4M mesh".

---

## Author Comment (AC2) · 16 Aug 2018

We would like to thank the referees for their valuable comments and appraisal. We are revising the manuscript in line with their comments. We have made significant additions to both the simulation software, as well as this paper, therefore we will be updating the software version number to 1.5, and we would also like to add two co-authors who have been instrumental in developing this new version.

[Figure]

**1 Referee 1**

Main points:

Point 1 – This is presumably the first model implementation of its kind, and I find that the implementation should be of interest for the tsunami community, as well as other scientific communities with interest of solving shallow water wave equations or related problems without in depth knowledge of different types of hardware architecture. This part is highly regarded.
Reply – We give a better overview of the model in the paper, however, the numerical implementation has been described in detail in previous work (which we now highlight better).

Point 2 – The study of the speedup on different types of hardware are also new, and the findings are interesting in their own right. However, if possible, I encourage the authors to see if it is possible to compare the model speedup also with other models (such as HySEA) for inter model comparison.
Reply – Direct comparison is very difficult, because the open source models and implementation available all differ from ours in various, but significant ways - e.g. Tsunami-HySEA only supports structured meshes (vs. our unstructured meshes) and a single GPU - support for nested meshes and multi-GPU are not available in the public version. Instead, we pull results from their published papers to discuss the performance and the speedups they achieve.

Point 3 – The validation of the model is entirely missing. I know that the previous VOLNA codes have been benchmarked towards NTHMP tests previously, but this is a new implementation. While one may expect a similar accuracy for this code as well, validation needs to be demonstrated. To emphasise this, the novelty of this
paper actually hinges on some kind of proof; i.e. that the model can produce results consistent with previous versions. Moreover, no explicit tsunami results are shown, only results showing the speedup. As a minimum, some results showing that the tsunami code gives a reasonable output needs to be included. I would propose that the authors include one or two of the standard tsunami inundation benchmark tests. I'm sure the authors have some such tests available.

Reply We are adding results of two NTHMP benchmarks to demonstrate the numerical accuracy of the code.

Point 4 – The text and reference list is a bit imbalanced with respect to the authors own work. It would be beneficial if some more external references are added, reference to external work is moved upfront, or alternatively, discussion of the authors own work that are not strictly relevant for this paper are omitted (some parts seems not strictly necessary, see below). In the line-by-line review below some examples are listed. The references in the related work section should be moved upfront (section 2 seems unnecessary).

Reply – We have included a considerable number of additional references to related work, but for better structure and readability we kept the Related Works section of the paper - as many publications in GMD do.

Line-by-line comments for edits to the text and references:

Comment on Page 1, line 14 – More references to external work should preferably be placed up front (e.g. here). It makes sense to pay attention to the general literature first, and use this to put the authors own work into a general context thereafter.

Reply – We are adding a number of citations, particularly to work related to tsunami simulation, to this first paragraph.

Comment on Page 1, line 15 – The statement "there are only a handful of codes that are suitable for integration into a workflow" is unsubstantiated, please remove.
Reply – We have removed this sentence

Comment on Page 2, line 1 – The science perspective is missing here, but is obvious, for instance the need for running sensitivity analysis (such as varibable slip or uncertainty assessments, e.g. Goda et al., 2014), probabilistic tsunami hazard assessments (e.g. Geist and Parsons, 2006; Davies et al., 2018; Grezio et al., 2017), or for more efficient and informed tsunami early warning (e.g. Oishi et al., 2015, Castro et al., 2015). I think it would strengthen the paper to mention and discuss such examples.
Reply – We have placed additional references in the introduction to emphasize the science perspective as the reviewer suggested.

Comment on Page 2, lines 19-28 – Reading this paragraph, you get the impression that the Volna code is unique with respect to workflow integration, which is not the case (see comment to Page 1, line 15). There are probably more than 10 codes worldwide that can do much of the same analysis. Granted, the new development presented in this paper provide new opportunities wrt hardware independence. This should be the main message.
Reply – We clarify that it is Volna-OP2's performance and portability that prompted its use in the cited papers and its integration into workflows.

Comment on Page 3 – References and discussions in Related Work section should preferably be moved upfront.
Reply – We have included a considerable number of additional references to related work, but for better structure and readability we kept the Related Works section of the paper - as many publications in GMD do.

Comment on Page 3, line 10 – Unsubstantiated statement: "Since there is no consensus as to their advantage. . .". What do the authors mean here? Clarify, or remove statement. Simply, Boussinesq models are needed wherever tsunami dispersion is needed (see e.g. Glimsdal et al., 2013), otherwise the shallow water approximation is sufficient.

Reply – We have changed the text to say these models are primarily needed for dispersion, and added the citation.

Comment on Page 3, line 17 – The authors should provide a literature search here and add more references, as GPU implementation of shallow water is under rapid development. As a minimum, the authors needs to add reference to the GPU SWE code by Brodtkorp et al. (2010) and the Boussinesq GPU code Celeris (Tavakkol and Lynett, 2017).

Reply – We have added text to reference the work by Brodtkorp et al., Tabakkol and Lynett, Acuna and Aoiki, Liang et. al.

Comment on Page 4 – I could not find anywhere from the discussion whether related OP2 applications have been performed for other (similar) applications of hyperbolic equation. If such implementations exists, it would be of interest to discuss their performance.

Reply – Other applications implemented in OP2 are either elliptic or not PDE solvers. We place a comment referencing three key papers describing performance of other OP2 applications.

Comment on Page 5, line 9 – Remove "very".

Reply – We have removed the word
Comment on Page 5, lines 7-12 – Treatments of shocks and breaking waves are probably the main reason FV are used, so this needs explicit mentioning.
Reply – This is now mentioned in the paragraph

Comment on Page 5, line 14 – Replace "megatsunami" with either "large tsunami" or "transoceanic tsunami".
Reply – This was replaced for transoceanic tsunami

Comment on Page 5, line 15 – Again, we refer to Glimsdal et al. (2013). Because frequency dispersion is a time dependent property, important of dispersion increases with time for a given initial condition, so it is not sufficient to refer to dispersion as weak just based on the properties at a given snapshot. The discussion here seems to merge the effect of dispersion on deep water waves and inundation, which are very different. Either the authors needs to clarify better, however, it would probably be better to omit this discussion here, and rather state that the present implementation is based on the non-linear shallow water model (you do not have to justify that dispersion is not included).
Reply – As suggested, we omit this discussion

Comment on Page 6, first paragraph – This repeated reference to applications of the code seems awkward as it is not needed in this context, beside, this is already discussed in the introduction.
Reply – We reorganise this paragraph, removing most references.

Comment on Page 9, lines 14-15 – See previous comment.
Reply – We have removed these references

Comment on Page 14, lines 29-31 – I cant see that this is more relevant than other and more general applications such as PTHA and tsunami early warning. As said, a more general discussion with references from a broader literature is needed.

Reply – We have removed this paragraph as it was indeed too specific and it is not a discussion/conclusion really

**2   Referee 2**

Comment 1 – Use of uniform triangular meshes as opposed to non-uniform ones. Am I right if I say that the most typical use case of VOLNA-OP2 is with non-uniform meshes? If so, why using uniform ones? The mesh can have a drastic impact on the achieved performance, due to different MPI partitioning, load balancing, effectiveness of mesh renumbering etc. Is this a weakness of the analysis? if not, why?

Reply – we are re-evaluating scaling performance using non-uniform meshes of the same area

Comment 2 – I know that with other OP2-based applications, traditionally, MPI+OpenMP has never really outperformed pure MPI (due to issues that have been extensively described in prior work), or at least not by a significant factor. Is the situation different with VOLNA? If so, can you say why? Speculation: is it because VOLNA is "more compute-bound" than other codes used in the past? I'm asking because I see that all numbers reported derive from MPI+OpenMP. By the way, I assume that you have taken care of pinning etc – can this be confirmed?

Reply – This is a good point, we added further performance figures to the paper and a brief discussion on the difference. MPI performs better at low node counts, but scales worse the MPI+OpenMP on the KNL. Pinning was done, through the built-in

mechanisms in the MPI distribution.

Comment 3 – Was data alignment enforced for directly accessed datasets ? I understand the same dataset can be accessed directly in one loop and indirectly in another loop, but perhaps there's hope to exploit some alignment in at least some of the memory-bounded loops?
Reply – Data alignment is enforced and the appropriate annotations are supplied in the generated code, although in practice it made no difference to the resulting performance

Comment 4 – Can you be more precise as to why it was not possible to vectorise 'applyFluxes'? Can you share more details about how the vectorisation of the other loops has been achieved – is it via auto-vectorisation or something else?
Reply – The details of the vectorised code generation are discussed in a previous paper - this is now appropriately cited. It was not working on the loop in question due to a compiler bug - this was reported to Intel, but not yet fixed.

Comment 5 – Which loops are compute-bound and which are memory-bound?
Reply – In this paper we did not really want to go into too much detail on the optimisations, as that may not really be of too much interest to the reader of this journal. We place short notes on which loops are bandwidth/compute/control limited. Detailed performance analysis of these loops individually is available in prior work, which we now cite: I. Z. Reguly, E. László, G. R. Mudalige, and M. B. Giles. 2014. Vectorizing Unstructured Mesh Computations for Many-core Architectures. In Proceedings of Programming Models and Applications on Multicores and Manycores (PMAM'14). ACM, New York, NY, USA, , Pages 39 , 12 pages. DOI=http://dx.doi.org/10.1145/2560683.2560686

Comment 6 – Can you report the max memory bandwidth of *all* platforms? I think some are currently missing. Also, how was the memory bandwidth limit determined ? specs, STREAM, or...?

Reply – We report on the max bandwidth achieved on different systems, as measured by STREAM Triad.

Comment 7 – Can you state the model of XeonPhi used? Some have 68 cores, not 64 like the one used in the experimentation.

Reply – We added a note to the text on using the 64 core 7210 model.

Comment 8 – I think comparing the VOLNA-OP2 performance in different architectures is a bit unfair, at least for two reasons: 1- The number of "degrees-of-freedom" (or simply triangles) per core is generally different, and so is the proportion of time spent is computation and communication. Even data locality may have been impacted. 2- These architectures are profoundly different among them – in theoretical attainable peak performance, price, release date, etc. I don't like sentences such as "The GPU system with X1 nodes was Y times faster than the CPU system with X2 nodes"; I don't think they add much to the paper, and in fact they may be misleading. Perhaps you can normalise over a common metric, but again that's quite tough. Also, the Phi suffers a lot from the lack of the vectorisation in 'applyFluxes', and it's unclear whether this issue can be worked around or not.

Reply – While for computer science benchmarking purposes we entirely agree that such a comparison is not fair, this paper is aimed at people who use and run tsunami modeling software (most of the authors are not computer scientists). We believe that for this audience such speedup numbers are relevant, as they will have to choose a platform to run on, and their basis for decision is only the cost of access and the relative performance between them. We dedicate a section to this (Running Costs and Power Consumption), and now point out that this kind of comparison is indeed unfair.

Comment 9 – In the conclusions, I disagree with the sentence: "Through performance scaling and analysis of the code on (...), we have demonstrated that VOLNA-OP2 indeed delivers near-optimal performance on a variety of platforms (...)". My point is that I don't think that you have demonstrated that you're achieving near-optimal performance – it is true that some loops are relatively close to the architecture memory bandwidth limit, but others are not. We don't know the reason – in fact, we don't even know whether these loops are compute- or memory-bound (see point above). So I suggest to either drop that sentence, or rephrase it, or add a roofline plot.
Reply – We have dropped this statement, especially because for unstructured mesh computations this is difficult to quantify. Also because fair comparison to other codes was not possible.

Comment 10 – Of course, I assume that comparing the performance of VOLNA-OP2 to that of other codes is way too difficult, if not impossible
Reply – Unfortunately we found no open-source codes that would allow direct comparison. We have extracted some performance figures from related papers

Comment 11 – I see that version 1.0 has been "Zenodoized" – don't forget to add the DOI to the paper. Also, I suggest to move the code to its own organisation on GitHub so that permissions can be more easily set and contributing to VOLNA-OP2 gets easier. I also suggest to link VOLNA-OP2 from the 'apps' section of OP2.
Reply – We have added DOIs to the paper

Comment 12 – It might be just me, but it feels like that different sections of the paper have been written by different people. That is, the transition from some paragraphs to others is not as smooth as it might be. Also, there are some typos (e.g., outline

-> outlined) and/or missing words ("the physical across") in various sections. All this should be improved prior to publication.

Reply – We are making changes to the text to make the transitions smoother, and clearing up typos.

Comment 13 – Section 3.1 might benefit from some figures. I know this is not the main point of the article, but I'm not sure that readers who are unfamiliar with OP2 will be able to understand how, for example, GPU parallelisation works. Maybe just cite some prior paper?

Reply – In the interest of brevity, we added citations to previous work detailing the parallelisation approaches.

Comment 14 – In the performance section, Table 1 gives a name to three meshes: M1, M2, M3. These could be used systematically throughout the whole section, and in the plots as well, instead of referring to "the 1.4M mesh".

Reply – We have made the suggested changes - with the new non-uniform meshes, there are labelled as NU_3...NU_0

---

## Author Response (AR3)

Dear Editor,

Thank you for the feedback! We have addressed the points of the minor revision as follows:

- The validation towards the NTHMP benchmark cases could in principle be dependent of the architecture they are implemented in. Therefore, one page 7 and 8, please provide a reference to which architecture that is used for the benchmark cases.

- Answer: we have added the description of the platforms used for these benchmarks (Page 8, line 14, page 9, line 22).

- Secondly, the same issue applies to the Makran demonstration case. Could the authors provide a comparison between two time series (such as Figure 8) also for all the different architectures tested, for the same grid resolution? Optimally, machine precision should be obtained.

- Answer: We add a sentence to the discussion of OP2 about potential differences between results obtained on different architectures (page 5, line 19-21) We also add a sentence on the specific differences on the Makran case to section 5.2 (page 13, line 27). The differences are so small that they would not be visible on a plot, therefore we just note the maximum relative difference (1.5%).

- Reference style needs to be fixed throughout (double parentheses are used, years are missing in several occasions).

- Answer: We have reviewed the text and corrected these issues (pages 1-4)

- Page 3 line 21: "Storkes" –¿ Stokes"

- Answer: We have corrected the typo (Page 3, line 26)

- Table 1: The number of computational nodes could preferably be added to the table.

- Answer: The table discusses details of the mesh - details of the architectures we run on, and the node counts are in the text of Sections 5.3, 5.4, 5.5

Yours sincerely,

Istvan Reguly et. al.

[revised manuscript text omitted]